# Embedding stochastic dynamics of the environment in spontaneous activity by prediction-based plasticity

**Toshitake Asabuki[1,2,3]\*, Claudia Clopath[1]\***

[1]Department of Bioengineering, Imperial College London, London, United Kingdom; [2]RIKEN Center for Brain Science, RIKEN ECL Research Unit, Wako, Japan; [3]RIKEN Pioneering Research Institute, Wako, Japan

## eLife Assessment

This is an **important** study that investigates how neural networks can learn to stochastically replay presented sequences of activity according to learned transition probabilities. The authors use error-based excitatory plasticity to minimize the difference between internally predicted activity and stimulus-driven activity, and inhibitory plasticity to maintain E-I balance. The approach is **solid** but the choice of learning rules and parameters is not always always justified, with some unclear aspects to the formal derivation.

**\*For correspondence:**
toshitake.asabuki@gmail.com (TA);
c.clopath@imperial.ac.uk (CC)

**Competing interest:** The authors declare that no competing interests exist.

**Abstract** The brain learns an internal model of the environment through sensory experiences, which is essential for high-level cognitive processes. Recent studies show that spontaneous activity reflects such a learned internal model. Although computational studies have proposed that Hebbian plasticity can learn the switching dynamics of replayed activities, it is still challenging to learn dynamic spontaneous activity that obeys the statistical properties of sensory experience. Here, we propose a pair of biologically plausible plasticity rules for excitatory and inhibitory synapses in a recurrent spiking neural network model to embed stochastic dynamics in spontaneous activity. The proposed synaptic plasticity rule for excitatory synapses seeks to minimize the discrepancy between stimulus-evoked and internally predicted activity, while inhibitory plasticity maintains the excitatory-inhibitory balance. We show that the spontaneous reactivation of cell assemblies follows the transition statistics of the model's evoked dynamics. We also demonstrate that simulations of our model can replicate recent experimental results of spontaneous activity in songbirds, suggesting that the proposed plasticity rule might underlie the mechanism by which animals learn internal models of the environment.

## Introduction

The brain is thought to use its sensory experience to learn an appropriate internal model of the environment, which can improve perception and behavioral performance. (*Merfeld et al., 1999*; *Lewald and Ehrenstein, 1998*; *Bell et al., 1997*; *Yasui and Young, 1975*; *Wolpert et al., 1995*). Such learning is thought to be fundamental to higher-order cognitive processes such as perception, decision making, and prediction of sensory stimuli. Recent computational and experimental evidence suggests that the brain's learned internal model may be reflected in spontaneous activity. For example, in the visual cortex of awake ferrets, spontaneous activity shows spatial similarity to activity elicited by natural scenes (*Berkes et al., 2011*). Furthermore, hippocampus generates sequential replay of place fields during rest and sleep (*Wilson and McNaughton, 1994*; *Skaggs and McNaughton, 1996*; *Lee and*

*Wilson, 2002*). Such hippocampal replay occurs in a highly stereotyped temporal order, with the same sequence of replayed activities often observed across multiple events (*Davidson et al., 2009*; *Diba and Buzsáki, 2007*; *Gupta et al., 2010*; *Wu and Foster, 2014*).

Several computational studies have proposed variants of Hebbian plasticity rules for learning deterministic or even stochastic switching dynamics of replayed activities (*Levy et al., 2001*; *Litwin-Kumar and Doiron, 2014*; *Triplett et al., 2018*; *Ocker and Doiron, 2019*; *Asabuki and Fukai, 2024*). However, it has been challenging to extend these results to generate dynamic spontaneous activity obeying appropriate transition probabilities learned through sensory experience. Finding a plasticity rule which is capable of learning structured transitions in spontaneous activity could be instrumental for understanding the mechanism underlying cognitive processes in the brain.

In this paper, we propose a local biologically plausible plasticity rule for learning the statistical transitions between assemblies in spontaneous activity. We use a recurrent spiking neural network model consisting of distinct excitatory and inhibitory populations. The proposed synaptic plasticity rule for excitatory synapses seeks to minimize the discrepancy between stimulus-evoked and internally predicted activity, while inhibitory plasticity maintains the excitatory-inhibitory balance. We explore the potential performance of our model by learning the Markovian transition statistics of evoked network states. Our results show that the trained model exhibits spontaneous stochastic transitions of cell assemblies, even after structured external inputs are removed. We show that the transition statistics of spontaneous activity show a striking similarity to those of the evoked dynamics.

To further validate our model, we compare the model behavior with recent experimental results in songbirds (*Bouchard and Brainard, 2016*), which show that the uncertainty of upcoming states in a bird song modulates the degree of neural predictability. Our model replicates this experimental result, suggesting that the connectivity structure learned via the proposed plasticity mechanism could plausibly underlie the songbird's learned internal model.

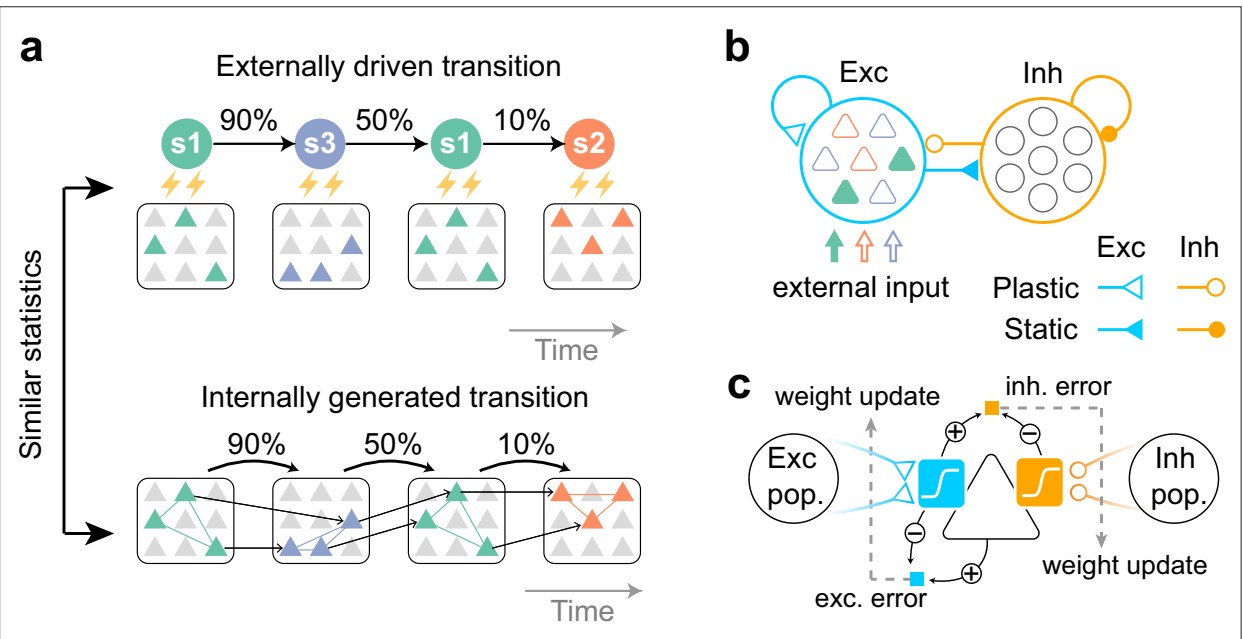

**Figure 1.** Task to be learned. (**a**, top) An example of a task used to test the model. Stimulus patterns evolve in time according to structured transition probabilities. The presentation of each stimulus pattern activates the corresponding group of neurons. Recurrent connections are learned by synaptic plasticity (**a**, bottom). The learned network should replay assemblies spontaneously, where the transition statistics are consistent with the evoked stimuli. (**b**) A network model with distinct excitatory and inhibitory populations. Only excitatory populations are driven by external inputs. Only synapses that project to excitatory neurons are assumed to be plastic. (**c**) A schematic of the proposed plasticity rules. Excitatory (blue) and inhibitory (orange) synapses projecting to an excitatory neuron (triangle) obey different plasticity rules. For excitatory synapses, errors between internally driven excitation (blue sigmoid) and the output of the cell provide feedback to the synapses (dashed arrow) and modulate plasticity (blue square; exc. error). All excitatory connections seek to minimize these errors. For inhibitory synapses, the error between internally driven excitation (blue sigmoid) and inhibition (orange sigmoid) must be minimized to maintain excitation-inhibition balance (orange square; inh. error).

# Results

## Spontaneous replay of learned stochastic sequences

While most studies have investigated plasticity mechanisms for learning random switching (*Litwin-Kumar and Doiron, 2014*; *Triplett et al., 2018*; *Ocker and Doiron, 2019*; *Asabuki and Fukai, 2024*) or deterministic transitions (*Chadwick et al., 2015*) between cell assemblies, our objective is to create a network model that spontaneously replays stochastic sequences of assemblies following synaptic plasticity. To that end, we first design a simple task whereby stimuli undergo stochastic transitions over time, and presentation of each stimulus increases excitatory drive to neurons targeted by that pattern (*Figure 1a,top*). We assume that a non-overlapping subset of excitatory network neurons receive its preferred stimulus (*Figure 1b*). After learning, the network should replay stochastic sequences of assemblies with transitions that are statistically consistent with evoked dynamics, without relying on external stimuli (*Figure 1a, bottom*).

We examined the possible learning mechanisms of stochastic neural sequences with a recurrent spiking network. Our network model consists of excitatory (E) and inhibitory (I) model neurons (*Figure 1b*). Only excitatory neurons are driven by external stochastic sequences. Initially, neurons in the network have random recurrent connections.

To learn a network model to obtain transition statistics of evoked dynamics, we proposed different local plasticity mechanisms for excitatory and inhibitory synapses. We assumed that only connections onto excitatory neurons were plastic (*Figure 1b*), while all others (i.e. connections onto inhibitory neurons) were fixed. In the excitatory recurrent connectivity, all synaptic weights were modified to reduce the error between internally generated and stimulus-evoked activities (*Figure 1c, blue square*). This plasticity rule is mathematically similar to that proposed in *Pfister et al., 2006*; *Urbanczik and Senn, 2014*, which minimizes the discrepancy between the somatic and dendritic activities (*Asabuki and Fukai, 2020*; *Asabuki and Fukai, 2024*). Through this process, excitatory synapses that contribute to predicting neural activity will be strengthened, thereby increasing the similarity between spontaneous and evoked activity. Instead of predicting the firing rate of neurons, the inhibitory synapses were modified to predict the recurrent excitatory potential (*Figure 1c, orange square*). This inhibitory plasticity is crucial for the network to maintain excitatory-inhibitory balance (*Vogels et al., 2011*) and generate spontaneous replay of stochastic assembly sequences, as we will see later. All feedforward connections were fixed and receptive fields were preconfigured. Finally, as in the previous study (*Asabuki and Fukai, 2024*), parameters of the response function are regulated according to the activity history of individual neurons (Methods). This regulation maintains the appropriate dynamic range of activities irrespective of the strength of external stimuli.

To examine how external stochastic sequences can influence network wiring, we trained a network model driven by stochastic external inputs. These inputs were generated by first-order Markovian chains with three 200ms long states, governed by fixed transition probabilities (*Figure 2a*). During training, excitatory synapses were modified much quicker than inhibitory synapses (*Figure 2b*). This difference in plasticity timescales follows from the nature of our learning rules: the wiring of excitatory synapses is reorganized by external stimuli, while inhibitory synapses only change to rebalance excitation. As such, excitatory plasticity in our model occurs before inhibitory plasticity, consistent with the experimental results (*D'amour and Froemke, 2015*). Indeed, even when the learning rate of inhibitory plasticity was twice that of excitatory plasticity, inhibitory plasticity still occurred on a slower timescale than excitatory plasticity (*Figure 2—figure supplement 1*).

We then asked how plasticity affects the neural dynamics by comparing the spontaneous activities of the network before and after learning. Here, we simulated spontaneous activity by replacing the temporally structured stimulation (i.e. the Markovian chain in *Figure 2a*) with constant background input. Furthermore, all synapses were kept fixed during spontaneous activity. Before learning, due to uniform initial connectivity, all excitatory neurons showed synchronous and spatially unstructured spontaneous activity (*Figure 2—figure supplement 2*). However, after learning, three cell assemblies emerged in the network, each of which encoded one external stimulus. Sequences of these cell assemblies were replayed stochastically in spontaneous activity (*Figure 2c*), and durations of given assembly reactivations were biased toward shorter durations but distributed broadly (*Figure 2d*).

We next asked whether or not the statistics of assembly switching were influenced by the temporal structure of the external sequence received by the network while it was learning. Since each assembly reactivation was contingent upon the previous assembly, statistics of external sequence may influence

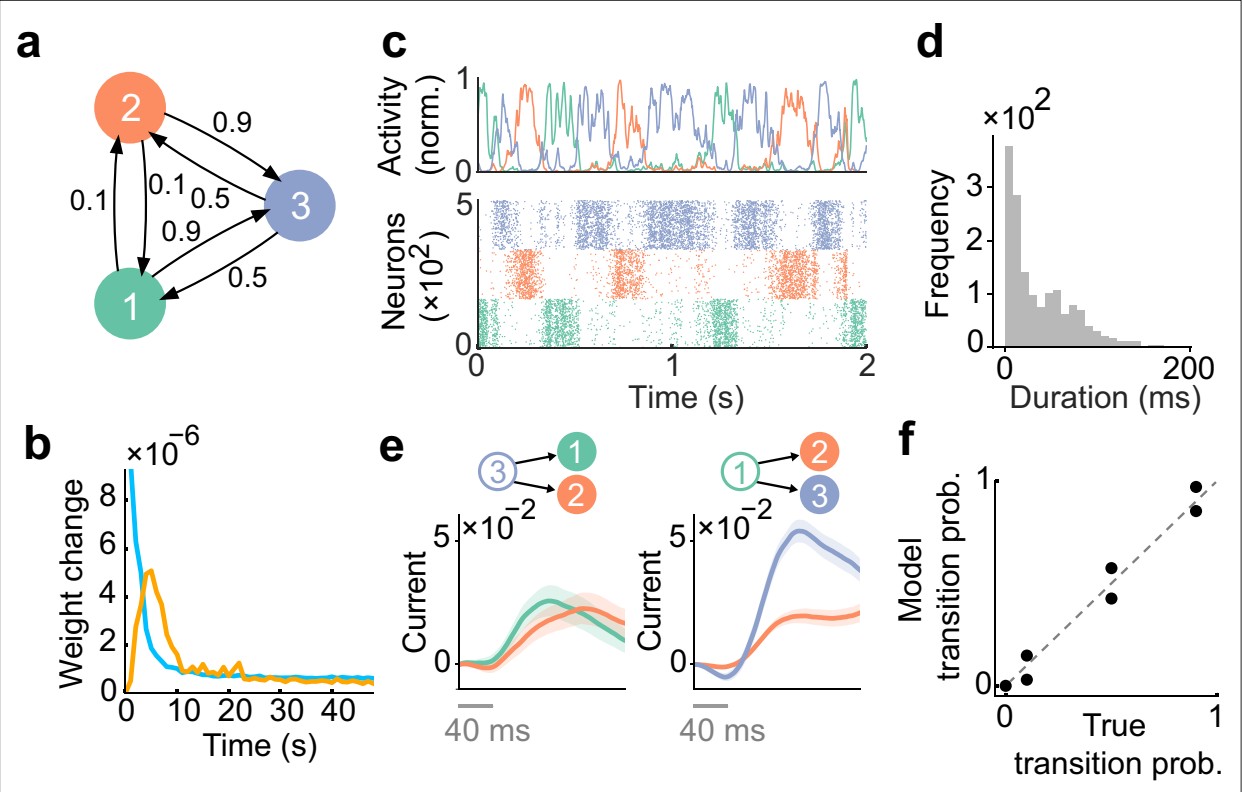

**Figure 2.** Spontaneous replay of stochastic transition of assemblies. (**a**) First, we considered a simple stochastic transition between three stimulus patterns. (**b**) Dynamics of weight change via plasticity. Excitatory synapses (blue) converged quicker than inhibitory synapses (orange). (**c**) Example spontaneous assembly reactivations (top) and raster plot (bottom) of the learned network are shown. Colors indicate the corresponding stimulus patterns shown in a. (**d**) Distribution of assembly reactivations. (**e**, left) The network currents to assembly 1 (green) and assembly 2 (orange) immediately after the reactivation of assembly 3 ceased. Both currents were similar in magnitude. (**e**, right) Currents to assembly 2 (orange) and assembly 3 (blue) immediately after the reactivation of assembly 1 ceased. The current to assembly 3 was stronger than that to assembly 2. (**f**) Relationship between the transition statistics of stimulus patterns and that of replayed assemblies. The spontaneous activity reproduced transition statistics of external stimulus patterns.

The online version of this article includes the following figure supplement(s) for figure 2:

**Figure supplement 1.** Inhibitory plasticity lags behind excitatory plasticity.

**Figure supplement 2.** Unstructured spontaneous activity before learning.

**Figure supplement 3.** The network performance is less sensitive to the duration of evoked assembly activations during learning.

**Figure supplement 4.** The network performance dependence on cell assembly size.

**Figure supplement 5.** The learning rate controls the duration of cell assembly reactivations.

**Figure supplement 6.** The network performance dependence on the strength of background input.

**Figure supplement 7.** The networks with more biologically plausible architectures.

the strength of synaptic currents via recurrent connectivity. To test this prediction, we first investigated how spontaneous reactivation of assembly 3 drives the subsequent assemblies (i.e. assemblies 1 and 2). Immediately after the reactivation of assembly 3 ceased, currents onto both subsequent assemblies increased gradually, without showing a significant difference (***Figure 2e, left***). This is due to the fact that state 1 and state 2 are structurally symmetrical in our setting (***Figure 2a***). We then asked how reactivation of assembly 1 drives the subsequent assemblies (i.e. assemblies 2 and 3). We note that the transition probabilities in the stimulus patterns were biased towards state 3 in this case (***Figure 2a***). Consistent with this bias between transition probabilities, we found that assembly 3 was driven much strongly than assembly 2 (***Figure 2e, right***). These results suggest that the temporal statistics of the trained external sequence influence the strength of synaptic currents that drive each assembly. We then quantified the similarity between the transition statistics of stimulus patterns and that of the

replayed assemblies. We defined the transition probabilities between assemblies by simply counting the occurrence of switching events over all possible pairs of assemblies (Methods). Comparison between transition probabilities of stimulus patterns and that of the reactivated assemblies revealed a clear alignment of temporal statistics (*Figure 2f*). Interestingly, even when the network was trained with input states of half the duration, the distributions of the durations of assembly reactivations remain almost identical to those in the original case (*Figure 2—figure supplement 3a*). Furthermore, the transition probabilities in the replay were still consistent with the true transition probabilities (*Figure 2—figure supplement 3b*).

We then asked how robust our model is to different stimulus settings and parameters. To this end, we first asked whether varying the size of the cell assemblies would affect learning. We ran simulations with two different configurations (in the task shown in *Figure 2*). The first configuration used three assemblies with a size ratio of 1:1.5:2. After training, these assemblies exhibited transition statistics that closely matched those of the evoked activity (*Figure 2—figure supplement 4a*). In contrast, the second configuration, which used a size ratio of 1:2:3, showed worse performance compared to the 1:1.5:2 case (*Figure 2—figure supplement 4b*). These results suggest that the model can learn appropriate transition statistics as long as the size ratio of the assemblies is not drastically varied. We next asked whether the speed of plasticity, controlled by the learning rate, would affect model performance. To see this, we trained the network model in two cases, one with a fast plasticity and one with a slow plasticity. We found that the two models still showed spontaneous assembly replay whose statistics clearly matched those of the evoked dynamics (*Figure 2—figure supplement 5b, d*). Interestingly, however, we found that the duration of assembly became longer in the slow learning case than in the fast case (*Figure 2—figure supplement 5a, c*). Finally, we found that the weaker background input causes spontaneous activity with a lower replay rate, which in turn leads to a high variance of the encoded transition (*Figure 2—figure supplement 6a, b*), while stronger inputs make the assembly replay transitions more uniform (*Figure 2—figure supplement 6c, d*).

We then tested whether learning performance would be affected by setting the ratio of excitatory to inhibitory neurons to 80% and 20% (*Figure 2—figure supplement 7a, left*). Even in such a scenario, the network still showed structured spontaneous activity (*Figure 2—figure supplement 7a, center*), with transition statistics of replayed events matching the true transition probabilities (*Figure 2—figure supplement 7a, right*). We then asked whether the model with our plasticity rule applied to all synapses would reproduce the corresponding stochastic transitions. We found the network could replay the appropriate transition only under some conditions (*Figure 2—figure supplement 7b*). The replay failed when the inhibitory neurons were no longer driven by the synaptic currents reflecting the stimulus, due to a tight balance of excitatory and inhibitory currents on the inhibitory neurons. We found similarly that when each stimulus pattern activates a non-overlapping subset of neurons, the network does not exhibit the correct stochastic transition of assembly reactivation (*Figure 2—figure supplement 7c*). Interestingly, when the activity of each neuron is triggered by multiple stimuli and has mixed selectivity, the reactivation reproduced the appropriate stochastic transitions (*Figure 2—figure supplement 7d*).

In summary, the plasticity rules in our model learn the transition statistics of evoked patterns while maintaining excitation-inhibition balance. Our results show that the prediction-based plasticity rule allows the model to learn and spontaneously replays the transition statistics of evoked patterns.

## Learned excitatory synapses encode transition statistics

To further understand the mechanism underlying the statistical similarity between the evoked patterns and spontaneous activity, we next asked how the transition statistics of stimulus patterns can influence network wiring. Over the course of training, the average weights of connections in each of the three cell assemblies increased gradually and converged to a strong value (*Figure 3a, middle and b, top*), indicating the formation of assemblies. On the other hand, we found that the average weights between each pair of assemblies decreased and settled at different stationary values (*Figure 3a, right and b, bottom*). After training, we reasoned that the transition probabilities between states should be encoded exclusively via between-assembly connections, as none of the states in the Markovian chain have self-transitions. To test this prediction, we first compared the average between-assembly connection matrix (*Figure 3a, right*) and the ground truth transition aligned well to the ground truth

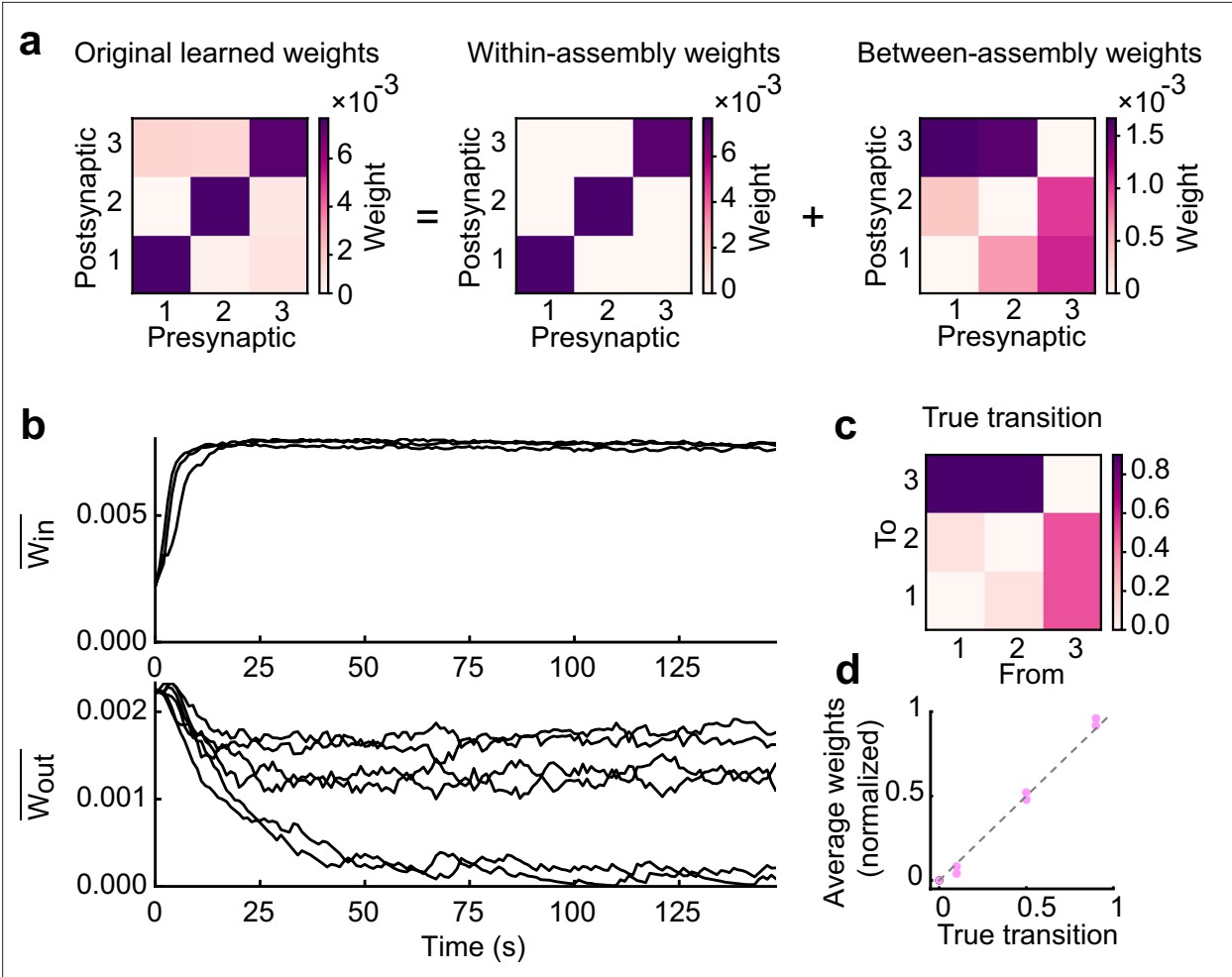

**Figure 3.** Learned excitatory synapses encode transition statistics. (**a**) A 3 by 3 matrix of excitatory connections, learned with the task in Fig.2a (left). The matrix can be decomposed to within- (middle) and between-assembly connections (right). (**b**) Strength of within- (top) and that of between-assembly excitatory synapses (bottom) during learning are shown. (**c**) True transition matrix of stimulus patterns. (**d**) Relationship between the strength of excitatory synapses between assemblies and true transition probabilities between patterns.

The online version of this article includes the following figure supplement(s) for figure 3:

**Figure supplement 1.** The role of inhibitory plasticity in transition probability learning.

**Figure supplement 2.** Network adaptation to task switching.

probabilities (*Figure 3d*). These results indicate that the network learns the temporal statistics of sequences by modifying the structure of inter-assembly excitatory connections.

The above analysis of excitatory weights revealed its crucial role in learning transition probabilities. Next, we examined the role of inhibitory plasticity in our model's function. To do so, we first simulated the network with fixed inhibitory weights performing the same task shown in *Figure 2*. We found that such a model exhibited spontaneous activity with blurred assembly structures compared to the original model (*Figure 3—figure supplement 1a*). Furthermore, the transition probabilities between replayed assemblies in this case did not show clear alignment with true transition (*Figure 3—figure supplement 1b*), though the excitatory weights reached values which did encode transitions (*Figure 3—figure supplement 1c, d*). These results suggest that maintenance of EI balance through inhibitory plasticity is necessary for generating structured spontaneous activity, even if excitatory connections learn transition probabilities.

## Network can adapt fast to task switching

In the above results, transitions between stimulus patterns obeyed fixed transition probabilities. We then wondered how the network learning would be affected if transition structures of stimulus patterns changed over time. To test such a scenario, we considered a case where the transition matrix in a Markovian chain switch between the first half and the second half of the learning phase (*Figure 3—figure supplement 2a*). We will refer these matrices as task1- and task2-matrix, respectively, and examine whether switching of transition matrixes influences the connectivity. During the first half of learning phase, between-assembly connections converged to certain values to encode task1-matrix (*Figure 3—figure supplement 2b, bottom*, 0–500 s). However, such stable connectivity reorganized quickly once the imposed task was switched to task2-matrix (*Figure 3—figure supplement 2b, bottom*, 500–1000 s). Note that in contrast to between-assemblies connections, within-assembly connections did not show such reorganization (*Figure 3—figure supplement 2b, top*). These results indicate that our model adapted to the second task even if distinct assembly structures were already formed during the first task.

To further understand how the model adapts to the new task, we next asked how error terms in excitatory and inhibitory plasticity (*Equations 11 and 13*) change through learning. As expected, the low-pass filtered errors $\mathrm{PE}^{exc}$ and $\mathrm{PE}^{inh}$ decreased as the network trained on task1 (*Figure 3—figure supplement 2c*, 0-500 s). However, once the task was switched, errors showed an abrupt increase followed by a gradual decrease as the network learned the second task (*Figure 3—figure supplement 2c* 500-1000 s; *Figure 3—figure supplement 2d*). Consistent with the previous result (*Figure 2b*), the peak of inhibitory error occurred delayed after that of excitatory one in each task (*D'amour and Froemke, 2015*; *Vogels et al., 2011*; *Figure 3—figure supplement 2d*). In summary, our model is also capable of task switching, via the reorganization of its weight structures through continuing plasticity.

## The network can learn complex stochastic sequences

So far, we have considered the capabilities of our model in regard to the relatively simple class of stochastic dynamics. In particular, the task we considered above contains only three states, and the transition structure was symmetric. In a realistic sequence, like the song of a bird, transition statistics are typically heterogeneous and more structured. To evaluate the model performance over a wide variety of structures, we now consider a transition diagram with more complex structure (*Figure 4a*). Despite its complex structure, the learned network showed spontaneous reactivations of all assemblies evoked during learning (*Figure 4b*), and the transition dynamics between these assemblies were governed by learned transition probabilities (*Figure 4c*). Indeed, the learned weight structures were consistent with the transition probabilities between states as we have seen in simpler task (*Figure 4—figure supplement 1*).

Recent experimental studies which examined temporal community structure (i.e. highly structured graph structure consisting of clusters of densely interconnected nodes; *Figure 4d*) found that human subjects tend to associate a given visual stimulus with other stimuli within the same 'community' (*Schapiro et al., 2013*; *Pudhiyidath et al., 2022*). To investigate whether the model can learn to associate states within a stimulated community, we first trained the network with a stochastic sequence of inputs, generated by a random walk over a graph with temporal community structure (*Figure 4d*). The learned model showed stochastic assembly transition during spontaneous activity (*Figure 4e*) relying on the appropriate weight structure (*Figure 4f*). Although transitions occurred between all pairs of assemblies, transitions between connected states in the diagram occurred much frequently than transitions between disconnected states (*Figure 4g*). This is because plasticity formed strong excitatory connections between assemblies with nonzero transition probabilities, as shown in *Figure 4f*.

In human participants, low-dimensional representations of evoked activities in different cortical regions have been reported to show clusters consistent with the structure of communities (*Schapiro et al., 2013*). To test whether our model could reproduce such representation of communities, we analyzed the low-dimensional representation of evoked activities in our model by applying principal component analysis (PCA) (see Methods). Such analysis revealed that the representations of stimulus patterns were grouped together into clusters or communities of mutually predictive stimuli, consistent with the experimental results (*Figure 4h*). We found that the clustered representations still exist even

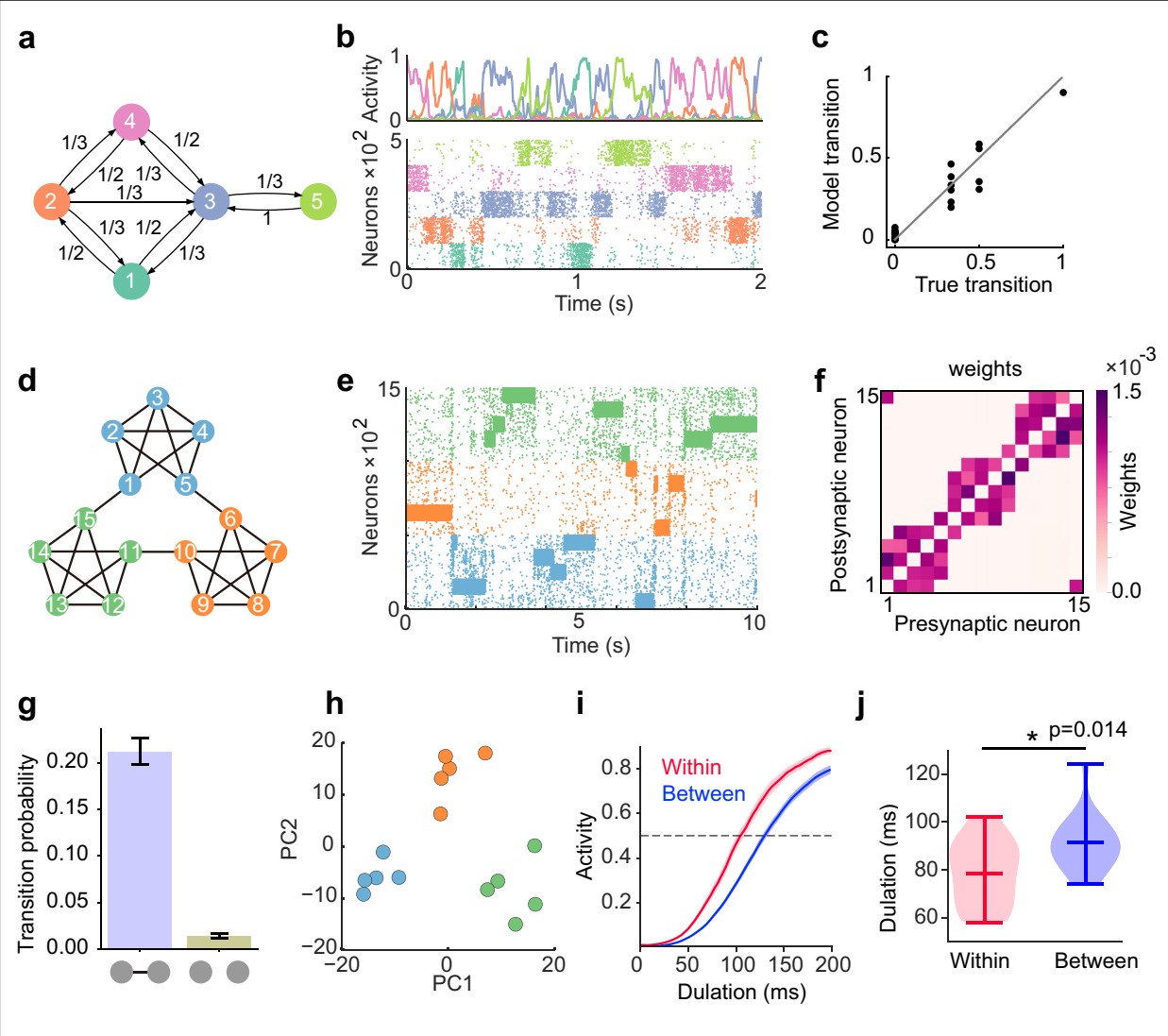

**Figure 4.** Learning complex structures. (**a**) Transition diagram of complex task. (**b**) Spontaneous activity of learned network. (**c**) Transition statistics of assemblies reproduce true statistics. (**d**) Transition diagram of temporal community structure. (**e**) Raster plot of spontaneous activity of the network trained over structure shown in (**d**). (**f**) Structure of learned excitatory synapses encode the community structure. (**g**) Spontaneous transition between assemblies connected in the diagram shown in d occurs much frequent than disconnected case. (**h**) Low dimensional representation of evoked activity patterns shows high similarity with community structure. (**i**) Time courses of replayed activities transitioning within (red) and between (blue) communities. (**j**) Comparison of mean durations in (**i**). P-value was calculated by two-sided Welch's t-test.

The online version of this article includes the following figure supplement(s) for figure 4:

**Figure supplement 1.** The excitatory synapses learned transition structures of complex task shown in *Figure 4a*.

**Figure supplement 2.** Low-dimensional community structure reflects learned weights, not input order.

if the input sequences were scrambled after learning (*Figure 4—figure supplement 2*), indicating that this result does not rely on the stimulus protocol, but instead on the learned weights.

We further asked whether within- and between-community reactivations showed any differences in terms of their behavior. To this end, we perturbed an assembly corresponding to non-boundary states in the first community (states 2–4 in the transition diagram shown in *Figure 4d*) and monitored the behavior of subsequent autonomous network activities. According to the above results, we expect that within-community reactivations should occur quicker than between-community assemblies, due to strong within-community coupling. To test this hypothesis, we calculated the duration from the end of the perturbation until subsequent activity reached a certain threshold (*Figure 4i*). As expected, the transition to within-community states showed much shorter durations than to between-community

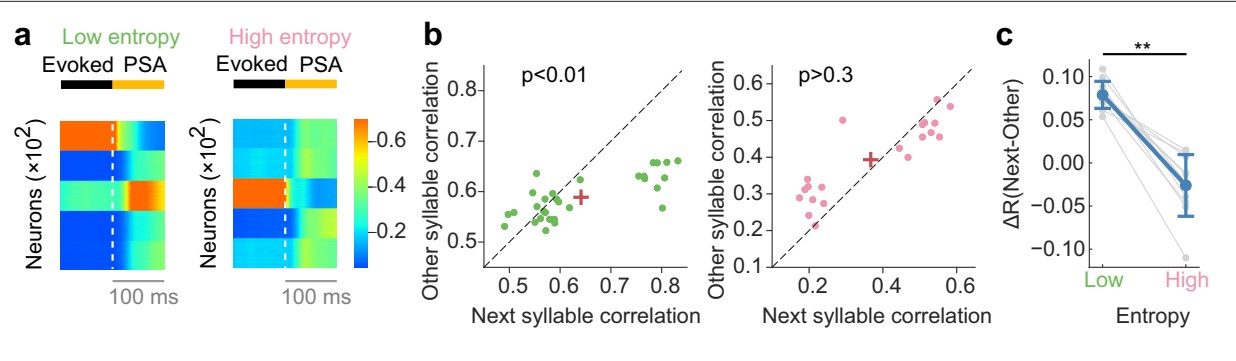

**Figure 5.** Network dynamics consistent with recorded neural data of songbird. (**a**) Example poststimulus activity (PSA) for low- (left) and high-entropy (right) transition cases. (**b**) Comparison of correlation coefficients between PSA and evoked single-syllable responses for next syllables and other syllables. For low entropy transition case, the next-syllables correlations were significantly higher than other-syllables correlations (p < 0.01, Wilcoxon signed-rank test) (left). In contrast, such correlation coefficients showed no significant difference for high entropy transition case (p > 0.3, Wilcoxon signed-rank test) (right). Red crosses are mean. (**c**) The difference in correlation coefficients between next and other syllables (ΔR) was significantly greater for low entropy transitions than for high entropy transitions (p < 0.01, two-sided Welch' s t-test).

case (*Figure 4j*), indicating that between-community transition occurred with much slower time scale compared to within-community case. Together, these results indicate that our network can learn complex temporal structures in spontaneous activity and reproduce the neural representation of the temporal community structure observed in the experiment.

## Network dynamics consistent with recorded neural data of songbirds

Finally, we tested whether the spontaneous activity in our model resembles recorded neural activity of HVC in Bengalese finch (Bf). Bf learns songs composed of multiple stereotyped short sequences, or syllables. The transitions between these syllables can be described via Markovian process with varying levels of certainty. Intuitively, given one syllable in a bird song, precise prediction about the neural response to the next syllable can be made if the transition from that syllable is highly certain, while imprecise transitions will lead to imprecise predictions about the neural response. Indeed, recent experimental study reported that uncertainty of upcoming syllables in a Bf song modulates the degree of predictability of subsequent neural activation (poststimulus activity; PSA) in HVC (*Bouchard and Brainard, 2016*). We sought to test whether our model would exhibit a similar property. To this end, we analyzed the behavior of a network model that had already learned the task (shown in *Figure 4a*). The transition structure we chose is relatively simple compared to the real song of a Bf, yet captures measured features of bird songs (i.e. both structures consist of highly certain and less-certain transitions). In the experiment, similarities were calculated between the trial-averaged PSA following a short sequence of stimuli, and the response to an isolated stimulus. To mimic this experimental design, we measured stimulus-triggered averages of our autonomous network activity as a proxy for PSA (*Figure 5a*). To examine how uncertainty of state transitions in a sequence influences predictive strength in network activity, we first calculated the Pearson correlation coefficient between PSA and responses to next states in a sequence. We will refer to such correlations as 'next-state correlations.' Note that if there were multiple next-states from a given state, all correlations corresponding to that state were averaged. We further calculated the correlation between PSA and responses to other states that did not follow the given state ('other-state correlations'). Similar to the next-state correlations, other-state correlations were averaged over all disconnected states from each state. We then compared next-state correlations and other-state correlations between highly certain (*Figure 5b, left*) and less-certain (*Figure 5b, right*) transitions. Here, highly certain transitions refer to those which have a transition probability greater than 1/2. Other transitions were classified as less-certain transitions. Consistent with experimental results, next-state correlations were significantly greater than other-state correlations in the highly certain case (*Figure 5b, left*). This correlation difference was less significant in less-certain case (*Figure 5b, right*). These results indicate that transition uncertainty modulated the degree to which PSA is predictive of upcoming states.

We performed a more direct comparison of predictive strength by measuring the difference between two types of correlations (i.e. next- and other-state correlations) over multiple levels of transition uncertainty. Here, for each state, next-state correlation was subtracted by other-state correlation. Transition uncertainties were quantified by calculating the conditional entropy of transition probabilities of stimulus patterns. Note that a higher value of entropy indicates less-certain transition, and vice versa. As expected, correlation differences increased as entropy decreased (*Figure 5c*), indicating that the predictive strength of network PSA was larger for low-entropy transitions (i.e. highly certain transitions) than for high-entropy transitions (i.e. less-certain transitions). What is the underlying mechanism of such predictability differences? Although each trial of assembly perturbation lead to subsequent reactivation of one of the assemblies, trial-averaged activities (i.e. PSAs) marginalized all possible transitions in the transition diagram (*Figure 5a*). Due to this averaging process, similarities between PSA and stimulus-evoked activities increase if conditional entropy is low (i.e. certain transition), and vice versa. Overall, our results suggest that our model learns transition statistics of stimulus patterns, with transition uncertainty influencing predictive strength in the network activity.

## Discussion

Understanding how the brain learns internal models of the environment is a challenging problem in neuroscience. In this study, we proposed synaptic plasticity rules for learning assembly transitions via sensory experiences. Our excitatory plasticity aims at minimizing the error between sensory-evoked and internally generated predictions of upcoming activity. We showed that the network learns the appropriate wiring patterns to encode the transition structure of states, and thus exhibits stochastic transitions between assemblies in spontaneous activity. We further showed that appropriate replay of stochastic transitions requires both excitatory and inhibitory plasticity. These plasticity rules showed a clear division of labor. For excitatory synapses, the connectivity learns transition probabilities during the evoked phase, and inhibitory plasticity seeks to maintain the excitatory-inhibitory balance. We showed that network excitatory plasticity alone cannot account for stochastic replay of learned activity, even if excitatory synapses learn an appropriate structure. Future experimental studies could examine our model's predictions by testing how blocking synaptic plasticity specifically in excitatory or inhibitory neuron populations distinctly impacts the transition statistics in spontaneous replay events.

Variants of the Hebbian plasticity rule have been widely used to learn the precise order of sequential reactivations. For example, a rate-based Hebbian rule has been proposed to generate trajectories along a chain of metastable attractors, each corresponding to a reactivation of a single network state (*Fonollosa et al., 2015*). Another proposed mechanism is that the transitions are governed by theta oscillations, which form a temporal backbone of the sequential reactivation of assemblies (*Chadwick et al., 2015*). Despite the successes of these Hebbian rules in learning precise order in sequences, plasticity rules that learn structured transition probabilities and replay them in spontaneous activity were still unknown.

How does our plasticity mechanism differ from the Hebbian rule? In the Hebbian rule, synaptic strength is potentiated as long as pre- and postsynaptic neurons show correlated activities. Due to the nature of the Hebbian rule, after sufficient potentiation, synapses reach a predefined upper limit, making the strength uniform among strong synapses (*Kempter et al., 1999*; *Song et al., 2000*; *Masquelier et al., 2008*). Such connectivity is useful when the network learns deterministic sequences, but it alone is insufficient to learn transition probabilities. In contrast, our proposed model aims at predicting the evoked activities by internally generated dynamics, so that learning ceases when the prediction error is sufficiently minimized. A similar plasticity rule has been proposed to minimize the discrepancy between stimulus-evoked and internally predicted activity, generating a stable synaptic distribution that allows pattern completion and unsupervised feature detection from noisy sensory input. (*Tavazoie, 2013*). These mechanisms result in learned synaptic distributions that are not uniform as observed in STDP, but rather converge to values proportional to the transition probabilities between assemblies (as shown in *Figure 3b*).

Our proposed plasticity mechanism could be implemented through somatodendritic interactions. Analogous to previous computational works (*Urbanczik and Senn, 2014*; *Asabuki and Fukai, 2020*; *Asabuki et al., 2022*), our model suggests that somatic responses may encode the stimulus-evoked neural activity states, while dendrites encode predictions based on recurrent dynamics that aim to minimize the discrepancy between somatic and dendritic activity. To directly test this hypothesis,

future experimental studies could simultaneously record from both somatic and dendritic compartments to investigate how they encode evoked responses and predictive signals during learning (*Francioni et al., 2023*).

The proposed mechanism of learning stochastic transitions between cell assemblies may offer several advantages over deterministic transitions, as suggested by previous studies. One possibility is that the internal dynamics of stochastic transitions can be used as prior knowledge about the structure of the world. In particular, the learned information about the transition statistics can be used to make probabilistic predictions about upcoming sensory events. It may also provide a flexible representation of the environment. In a deterministic case, assemblies are replayed in a fixed temporal order, which may make the network susceptible to noise or unexpected changes in the environment. In contrast, stochastic transitions may allow the network to generate rich repertoires of representations that could provide flexible computation against an uncertain environment.

In reinforcement learning (RL), balancing the tradeoff between exploration and exploitation to maximize a long-term reward signal is one of the most challenging problems. While both exploration and exploitation phases are crucial in RL, exploration is often much more difficult. This difficulty arises from the fact that exploration is especially important when the agent does not have an optimal policy. One way in which the agent might bypass or speed up this exploration phase is through prior knowledge of the environment's transition statistics. Furthermore, learning transition statistics as an internal model may be beneficial when an agent solves a task in an environment where the reward distribution is sparse. Having an internal model of the transition statistics may allow an agent to predict the expected value of the future reward for taking a particular action in a given state. However, the relationship between the reward-based plasticity rule and our proposed rule still needs further study.

Several computational models have demonstrated that Hebbian-like plasticity rule can learn appropriate Markovian statistics (*Kappel et al., 2014*; *Barber, 2002*; *Barber and Agakov, 2002*). However, our model differs conceptually from these previous models in some respects. While Kappel et al. demonstrated that STDP in winner-take-all circuits can approximate online learning of hidden Markov models (HMMs), a key distinction from our model is that their neural representations acquire deterministic sequential activations, rather than exhibiting stochastic transitions governing Markovian dynamics. Specifically, in their model, the neural representation of state B would be different in the sequences ABC and CBA, resulting in distinct deterministic representations like ABC and C'B'A', where 'A' and 'A' are represented by different neural states (e.g. activations of different cell assemblies). In contrast, our network learns to generate stochastically transitioning cell assemblies that replay Markovian trajectories of spontaneous activity obeying the learned transition probabilities between neural representations of states. For example, starting from reactivation from assembly 'A,' there may be an 80% probability to transition to assembly 'B' and 20% to 'C.' Although Kappel et al.'s model successfully solves HMMs, their neural representations do not themselves stochastically transition between states according to the learned model. Similar to Kappel et al.'s model, while the models proposed in *Barber, 2002* and *Barber and Agakov, 2002* learn the Markovian statistics, these models learned a static spatiotemporal input patterns only and how assemblies of neurons show stochastic transition in spontaneous activity has been still unclear. In contrast with these models, our model captures the probabilistic neural state trajectories, allowing spontaneous replay of experienced sequences with stochastic dynamics matching the learned environmental statistics.

Our model results were also compared to experimental results of sequence predictability in a songbird. Recent experiments have shown that the predictive uncertainty of the upcoming stimulus modulates the degree of similarity between stimulus-evoked and post-stimulus autonomous activity in the HVC of the Bengal finch (*Bouchard and Brainard, 2016*). However, the underlying mechanism is still unknown. Here, we have shown that a stochastic state transition in spontaneous activity can explain such a dependence of activity similarity on stimulus uncertainty. Our model predicts that the PSA reflects a trial average of stochastic transitions of evoked activity from a given stimulus. Trial-averaged neural activity washes out the variability of all possible realizations of the stochastic transition. Thus, PSA of an uncertain stimulus results in a combination of multiple transitions, leading to activity less similar than that evoked by a single stimulus. Several studies have shown that Hidden Markov Models or other statistical methods could account for the transition statistics in bird song (*Kogan and Margoliash, 1998*; *Katahira et al., 2011*). However, our study suggests that trial averaging operations can influence the degree of similarity between stimulus-evoked and post-stimulus activity.

Although we have shown that the proposed model can learn Markovian transitions, several studies suggest that animals often exhibit behaviors with non-Markovian or hierarchical statistics (*Seeds et al., 2014*; *Berman et al., 2016*; *Jovanic et al., 2016*; *Jin and Costa, 2015*; *Geddes et al., 2018*; *Markowitz et al., 2018*; *Kato et al., 2015*; *Kaplan et al., 2020*). In principle, our learning rule cannot be applied to learning non-Markovian transitions, since it only learns local transitions between states (*Brea et al., 2013*). Therefore, to learn higher-order stochastic transitions, recurrent neural networks like ours may need to integrate higher-order inputs with longer time scales. Another limitation of our model is that it cannot learn transition statistics if the states are separated in time. Both of these problems could be solved by considering working memory (WM) (*Baddeley, 1992*; *Miller and Cohen, 2001*) in an activity-dependent (*Funahashi et al., 1989*; *Goldman-Rakic, 1995*; *Fuster and Alexander, 1971*; *Amit and Brunel, 1997*) or activity-silent manner (*Mongillo et al., 2008*; *Barak and Tsodyks, 2014*; *Zucker and Regehr, 2002*; *Erickson et al., 2010*). Clarifying the relationship between the proposed prediction-based plasticity rule and plasticity rules that support memory traces, such as short-term plasticity, will warrant future computational studies.

Our work sheds light on the learning mechanism of the brain's internal model, which is a crucial step towards a better understanding of the role of spontaneous activity as an internal generative model of stochastic processes in complex environments.

## Methods

### Neural network model

Our recurrent neural networks consist of $N_E$ excitatory and $N_I$ inhibitory neurons. During learning, the membrane potentials of neurons at time $t$ with external current $I_i^{\text{ext}}$ were calculated as follows:

$$u_i^E(t) = \sum_{j=1}^{N_E} W_{ij}^{EE} x_j^E(t) - \sum_{k=1}^{N_I} W_{ik}^{EI} x_k^I(t) + I_i^{\text{ext}}(t), \tag{1}$$

$$u_i^I(t) = \sum_{j=1}^{N_E} W_{ij}^{IE} x_j^E(t) - \sum_{k=1}^{N_I} W_{ik}^{II} x_k^I(t), \tag{2}$$

where $u_i^E$ and $u_i^I$ are the membrane potential of $i$-th excitatory and inhibitory neuron, respectively (see *Table 1* for the list of variables and functions). The strength of external input $I_i^{\text{ext}}$ takes the value 1 if stimulus pattern targets neuron $i$ was presented and 0 otherwise. This structured external input was

**Table 1.** Definition of variables and functions.

| | |
|---|---|
| $u_i^E$, $u_i^I$ | Membrane potentials |
| $x_j^E$, $x_k^I$ | Postsynaptic potentials |
| $X_i^a$ | Poisson spike train generated by network neurons |
| $W_{ij}^{EE}$, $W_{ik}^{EI}$, $W_{ij}^{IE}$, $W_{ik}^{II}$ | Recurrent connections |
| $I_i^{\text{ext}}$ | External current elicited by stimulus presentation |
| $I_i^E$, $I_i^I$ | Synaptic currents generated by network neurons |
| $f_i^E$, $f_i^I$ | Instantaneous firing rates |
| $y_i^E$, $y_i^I$ | Recurrent predictions |
| $h_i$ | Memory trace |
| $\varphi$ | Dynamic sigmoidal function |
| $\bar{\varphi}$ | Static sigmoidal function |
| $\text{PE}^{\text{exc}}$, $\text{PE}^{\text{inh}}$ | Filtered prediction errors |

replaced to constant inputs $I_i^{\text{const}}$ of value 0.3 during spontaneous activity. We will describe the details of stimulus patterns later. $W_{ij}^{ab}$ $(a, b = E; I)$ is a recurrent connection weight from $j$-th neuron in population $b$ to $i$-th neuron in population $a$. All neurons were connected with a coupling probability of p=0.5. Initial value of synaptic weights $W_{ij}^{ab}$ were uniformly set to $0.1/\sqrt{pN_b}$ if $a = E$ and $1/\sqrt{pN_b}$ if $a = I$. $x_i^a$ is a postsynaptic potential evoked by $i$-th neuron in population $a$, which will be described later.

Spiking of each neuron model in population $E$ was modeled as an inhomogeneous Poisson process with instantaneous firing rate $f_i^E$ with a dynamic sigmoidal response function $\varphi$ with parameters of slope $\beta$ and threshold $\theta$ as:

$$f_i^E = \varphi\left(u_i^E; h_i\right) \equiv \varphi_0 \left[1 + \exp\left[g\beta\left(h_i\right)\left(-u_i^E + g\theta\left(h_i\right)\right)\right]\right]^{-1},$$
(3)

where $\varphi_0$ is the maximum instantaneous firing rate of 50 Hz and $g = 2$. The slope $\beta$ and threshold $\theta$ of sigmoidal function of population $E$ was regulated by the memory trace $h_i$ as:

$$\beta\left(h_i\right) = h_i^{-1}\beta_0$$
(4)

$$\theta\left(h_i\right) = h_i\theta_0,$$
(5)

where the values of constant parameters are $\beta_0 = 5$ and $\theta_0 = 1$. The memory trace tracks the maximum value of the short history of membrane potential $u_i^E$ as

$$\begin{aligned} \dot{h}_i &= -\tau_h^{-1}h_i, \quad \text{if } h_i > u_i^E \\ h_i &\leftarrow u_i^E, \quad\quad \text{otherwise} \end{aligned}$$
(6)

where $\tau_h = 10$ s is a time scale of memory trace. In the previous study (**Asabuki and Fukai, 2024**), such dynamic response function was introduced to prevent trivial solutions during the learning of recurrent and feedforward connections. In the current model, we assumed that each stimulus presentation drives a specific subset of network neurons with a fixed input strength, which avoids convergence to trivial solutions. Nevertheless, the dynamic sigmoid function could facilitate stable replay by regulating neuron activity to prevent saturation.

Inhibitory neurons' firing rate were assumed to be calculated with static sigmoidal function as:

$$f_i^I = \hat{\varphi}(u_i^I) \equiv \varphi_0 \left[1 + \exp\left(\beta_0(-u_i^I + \theta_0)\right)\right]^{-1},$$
(7)

Where the maximum instantaneous firing rate $\varphi_0$ was assumed to be same with that of excitatory neurons (i.e. 50 Hz). The parameters $\beta_0$ and $\theta_0$ are the constant values already appeared in *Equations 4 and 5*.

Neuron $i$ in population $a$ generates a Poisson spike train at the instantaneous firing rate of $f_i^a$. Let us describe the generated Poisson spike trains as:

$$X_i^a\left(t\right) = \sum_{t' \in t_i^a} \delta\left(t - t'\right),$$
(8)

where $\delta$ is the Dirac's delta function and $t_i^a$ is the set of time of the spikes of the neuron. The postsynaptic potential evoked by the neuron (i.e. $x_i^a$) was then calculated as:

$$\tau_s \dot{I}_i^a = -I_i^a + \frac{1}{\tau}X_i^a$$
(9)

$$\dot{x}_i^a = -\frac{x_i^a}{\tau} + x_0 I_i^a,$$
(10)

where $\tau_s = 5$ ms, $\tau = 15$ ms, and $x_0 = 25$.

## The learning rules

All excitatory synaptic connections onto excitatory neurons were modified to minimize the following cost function:

$$\mathfrak{L}_E = \sum_{i=1}^{N_E} \left[ f_i^E - y_i^E \right]^2, \tag{11}$$

where $y_i^E$ is a recurrent prediction of a firing rate, defined as:

$$y_i^E = \hat{\phi} \left( \sum_{j=1}^{N_E} W_{ij}^{EE} \cdot x_j^E \right), \tag{12}$$

where the function $\phi(.)$ is the static sigmoid function defined in *Equation 7*. The above cost function evaluates to what extent the recurrent potential predicts the activity of postsynaptic neurons. Taking the gradient of the cost function in *Equation 11*, we derived the plasticity rule for the excitatory plasticity as:

$$\Delta W_{ij}^{EE} \propto -\frac{\partial L_E}{\partial W_{ij}^{EE}}$$

$$\propto \left[ f_i^E - y_i^E \right] \cdot \frac{\partial y_i^E}{\partial W_{ij}^{EE}}$$

$$\propto y_i^E (1 - y_i^E) \cdot \left[ f_i^E - y_i^E \right] \cdot x_j^E. \tag{13}$$

While the term the term $y_i^E \left( 1 - y_i^E \right)$, which arose from the derivative of $y_i^E$, avoids saturation of neural activity, we show numerically this can be ruled out in the learning rule. Hence, the resultant plasticity rule for the excitatory synapses can be written as:

$$\Delta W_{ij}^{EE} = \epsilon \left[ f_i^E - y_i^E \right] \cdot x_j^E, \tag{14}$$

where $\epsilon$ is a learning rate and was set to $\epsilon = 10^{-4}$, unless otherwise specified.

We note that for Poisson spiking neurons, the derived learning rule is equivalent to the one that minimizes the Kullback-Leibler divergence between the distributions of output firing and the dendritic prediction, in our case, the recurrent prediction (*Asabuki and Fukai, 2020*). Thus, the rule suggests that the recurrent prediction learns the statistical model of the evoked activity, which in turn allows the network to reproduce the learned transition statistics.

Similarly, we defined the cost function for the inhibitory plasticity as:

$$\mathfrak{L}_I = \sum_{i=1}^{N_E} \left[ y_i^E - y_i^I \right]^2, \tag{15}$$

where $y_i^I$ was the total inhibitory input onto postsynaptic neuron:

$$y_i^I = \hat{\phi} \left( \sum_{j=1}^{N_I} W_{ij}^{EI} \cdot x_j^I \right). \tag{16}$$

Again, by taking the gradient with respect to $W_{ij}^{EI}$ derive the following inhibitory plasticity rule to maintain excitatory-inhibitory balance in all excitatory neurons:

$$\Delta W_{ij}^{EI} \propto -\frac{\partial L_I}{\partial W_{ij}^{EI}}$$

$$\propto \left[ y_i^E - y_i^I \right] \times \frac{\partial y_i^I}{\partial W_{ij}^{EI}}$$

$$\propto y_i^I (1 - y_i^I) \cdot \left[ y_i^E - y_i^I \right] \cdot x_j^I. \tag{17}$$

**Table 2.** Parameter settings.

| $p$ | Connection probability | 0.5 |
|---|---|---|
| $g$ | Gain parameter in sigmoid function | 2 |
| $N_E$, $N_I$ | Network size | 500, 500 (1500,1500 in **Figure 5e-j**; 800, 200 in **Figure 2—figure supplement 7**) |
| $\epsilon$ | Learning rate | $10^{-4}$ |
| $\tau_s$ | Synaptic time constant | 5 ms |
| $\tau$ | Membrane time constant | 15 ms |
| $\beta_0$, $\theta_0$ | Parameters for sigmoid | 5, 1 |
| $\tau_h$ | Time constant of memory trace | 10 s |
| $\varphi_0$ | Maximal firing rate | 50 Hz |
| $x_0$ | Scaling factor of synaptic current | 25 |
| $\tau_{avg}$ | Time constant for low-pass filtering the error | 30 s |
| $I_i^{const}$ | Constant external current during spontaneous activity | 0.3 |

In all simulations in this paper, we dropped the term $y_i^I \left(1 - y_i^I\right)$ and modified the inhibitory synapses according to the following rule:

$$\Delta W_{ij}^{EI} = \epsilon \left[y_i^E - y_i^I\right] x_j^I. \tag{18}$$

## Simulation details

The parameters used in the simulations are summarized in **Table 2**. All simulations were performed in customized Python3 code written by TA with numpy 1.17.3 and scipy 0.18. Differential equations were numerically integrated using an Euler method with integration time steps of 1 ms.

## Stimulation protocols

In all simulations, each stimulus patterns had a duration of 200 ms and were presented without inter-pattern interval. We assumed each neuron in a network was stimulated by one of stimulus patterns and targeted assemblies were not overlapped. Presentation of each pattern triggers excitatory current to its targeted neurons of strength 1 and zero otherwise. During spontaneous activity, stimulus patterns were replaced with constant background input $I_i^{const}$ for all excitatory neurons. In **Figure 5**, we assumed all excitatory neurons receive both structured and constant background inputs over whole period.

## Calculation of transition probabilities in spontaneous activity

In **Figures 2f, 4c and g**, we first calculated the population average of the instantaneous firing rates of all neurons in each assembly, during spontaneous activity. We will term such activities as assembly activities. We then defined the assembly reactivations by events that the assembly activities exceeded the threshold of which the value 50% of maximum value of each assembly activity. Transition probabilities between assemblies across all possible pairs were then calculated by counting the occurrences of reactivation of the subsequent assembly within 100ms of the end time of reactivation of the preceding assembly. In **Figure 2d**, durations of each assembly reactivation event were defined as a period during each assembly activation exceeded threshold.

## Calculation of weight changes

In **Figure 2b**, the weight changes were calculated every 2 s for excitatory and inhibitory synapses as:

$$\Delta W^{EE}(t) := \frac{\sqrt{\sum_{i,j} \left[W_{ij}^{EE}(t) - W_{ij}^{EE}(t - dt)\right]^2}}{N_E^2} \tag{19}$$

$$\Delta W^{EI}(t) := \frac{\sqrt{\sum_{i,j}\left[W_{ij}^{EI}(t) - W_{ij}^{EI}(t-dt)\right]^2}}{N_E N_I} \tag{20}$$

where $W_{ij}^{Ea}(t)$ $(a = E; I)$ is a synapse at time $t$ and $dt$ is a simulation time step of 1ms.

## Calculation of error dynamics in task switching

In *Figure 3—figure supplement 2c and 2d*, two types of prediction errors for excitatory and inhibitory plasticity were calculated as follows. First, we obtained the low-pass filtered errors $\mathcal{E}_i^{\mathrm{exc}}$ and $\mathcal{E}_i^{\mathrm{inh}}$ calculated by instantaneous error values in the plasticity rules (i.e. *Equations 14 and 18*) as:

$$\tau_{\mathrm{avg}}\dot{\mathcal{E}}_i^{\mathrm{exc}} = -\mathcal{E}_i^{\mathrm{exc}} + \left[f_i^E(t) - y_i^E(t)\right] \tag{21}$$

$$\tau_{\mathrm{avg}}\dot{\mathcal{E}}_i^{\mathrm{inh}} = -\mathcal{E}_i^{\mathrm{inh}} + \left[y_i^E(t) - y_i^I(t)\right], \tag{22}$$

where $\tau_{\mathrm{avg}} = 30$ s is a time constant for low-pass filter and $i$ is a neuron index. We then calculated the averaged errors $\mathrm{PE}^{\mathrm{exc}}$ and $\mathrm{PE}^{\mathrm{inh}}$ as:

$$\mathrm{PE}^{\mathrm{exc}}(t) = \frac{1}{N_E}\sum_{i=1}^{N_E}\left|\varepsilon_i^{\mathrm{exc}}(t)\right| \tag{23}$$

$$\mathrm{PE}^{\mathrm{inh}}(t) = \frac{1}{N_E}\sum_{i=1}^{N_E}\left|\varepsilon_i^{\mathrm{inh}}(t)\right|, \tag{24}$$

where $|\cdot|$ is an absolute value.

## Analysis of the low-dimensional representation in network

In *Figure 4h*, we first obtained matrix of network responses $U = (r_1, \ldots, r_{15})$, where $r_i$ $(i = 1, \ldots, 15)$ is a trial-averaged response of a whole network to one of 15 stimulus patterns shown in *Figure 4d*. Trial averaging was performed over multiple presentations of each stimulus. We then applied the PCA to matrix $U$ and visualized the low-dimensional representation of multiple stimulus in the learned network.

## Correlation measure for comparison with a songbird

In *Figure 5*, we calculated stimulus-triggered averages of autonomous network activity to obtain poststimulus activity (PSA) of a network model. In *Figure 5b and c*, the correlation between PSA and evoked activity triggered by one stimulus pattern was calculated neuron-wise and then averaged over all neurons.

## Acknowledgements

This work was supported by BBSRC (BB/N013956/1), Wellcome Trust (200790/Z/16/Z), the Simons Foundation (564408), and EPSRC (EP/R035806/1). The authors also thank Ian Cone for his comments on the manuscript and technical assistance.

## Additional information

### Funding

| Funder | Grant reference number | Author |
| --- | --- | --- |
| Biotechnology and Biological Sciences Research Council | BB/N013956/1 | Claudia Clopath |
| Wellcome Trust | 10.35802/200790 | Claudia Clopath |
| Simons Foundation | 564408 | Claudia Clopath |

| Funder | Grant reference number | Author |
| --- | --- | --- |
| Engineering and Physical Sciences Research Council | EP/R035806/1 | Claudia Clopath |

The funders had no role in study design, data collection and interpretation, or the decision to submit the work for publication. For the purpose of Open Access, the authors have applied a CC BY public copyright license to any Author Accepted Manuscript version arising from this submission.

## Author contributions

Toshitake Asabuki, Conceptualization, Data curation, Software, Formal analysis, Validation, Investigation, Visualization, Methodology, Writing – original draft, Writing – review and editing; Claudia Clopath, Conceptualization, Supervision, Funding acquisition, Writing – original draft, Writing – review and editing

## Author ORCIDs

Toshitake Asabuki ⓘ https://orcid.org/0000-0003-0951-5791
Claudia Clopath ⓘ https://orcid.org/0000-0003-4507-8648

Reviewer #2 (Public review): https://doi.org/10.7554/eLife.95243.3.sa1
Reviewer #3 (Public review): https://doi.org/10.7554/eLife.95243.3.sa2
Author response https://doi.org/10.7554/eLife.95243.3.sa3

# Additional files

## Supplementary files

MDAR checklist

## Data availability

The current manuscript is a computational study, so no data have been generated for this manuscript. Code is provided on the GitHub repository (https://github.com/TAsabuki/stochastic_transition; copy archived at *Asabuki, 2024*).

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
