## [Editor Report · eLife Assessment]

This is an **important** study that investigates how neural networks can learn to stochastically replay presented sequences of activity according to learned transition probabilities. The authors use error-based excitatory plasticity to minimize the difference between internally predicted activity and stimulus-driven activity, and inhibitory plasticity to maintain E-I balance. The approach is **solid** but the choice of learning rules and parameters is not always always justified, with some unclear aspects to the formal derivation.

---

## [Referee Report · Reviewer #2 (Public review)]

Summary:

This work proposes a synaptic plasticity rule which explains the generation of learned stochastic dynamics during spontaneous activity. The proposed plasticity rule assumes that excitatory synapses seek to minimize the difference between the internal predicted activity and stimulus-evoked activity, and inhibitory synapses try to maintain the E-I balance by matching the excitatory activity. By implementing this plasticity rule in a spiking recurrent neural network, the authors show that the state-transition statistics of spontaneous excitatory activity agrees with that of the learned stimulus patterns, which is reflected in the learned excitatory synaptic weights. The authors further demonstrate that inhibitory connections contribute to well-defined state-transitions matching the transition patterns evoked by the stimulus. Finally, they show that this mechanism can be expanded to more complex state-transition structures including songbird neural data.

Strengths:

This study makes an important contribution to computational neuroscience, by proposing a possible synaptic plasticity mechanism underlying spontaneous generations of learned stochastic state-switching dynamics that are experimentally observed in the visual cortex and hippocampus. This work is also very clearly presented and well-written, and the authors conducted comprehensive simulations testing multiple hypotheses. Overall, I believe this is a well-conducted study providing interesting and novel aspects on the capacity of recurrent spiking neural networks with local synaptic plasticity.

Weaknesses:

This study is very well-thought out and theoretically valuable to the neuroscience community, and I think the main weaknesses are in regard to how much biological realism is taken into account. For example, the proposed model assumes that only synapses targeting excitatory neurons are plastic, and uses an equal number of excitatory and inhibitory neurons.

The model also assumes Markovian state dynamics while biological systems can depend more on history. This limitation, however, is acknowledged in the Discussion.

Finally, to simulate spontaneous activity, the authors use a constant input of 0.3 throughout the study. Different amplitudes of constant input may correspond to different internal states, so it will be more convincing if the authors test the model with varying amplitudes of constant inputs.

Comments on revisions:

The authors have addressed all of the previously raised concerns satisfactorily, by running extra simulations with a biologically plausible composition of excitatory and inhibitory neurons, plasticity assumed for all synapses, and varied amounts of constant inputs representing internal states or background activities. While in some of these cases the stochastic dynamics during spontaneous activity change or do not replicate those of the learned stimulus patterns as well as before, these extended studies provide thorough evaluations of the strengths and limitations of the proposed plasticity rule as the underlying mechanism of stochastic dynamics during spontaneous activity. Overall, the revision has strengthened the paper significantly.

---

## [Referee Report · Reviewer #3 (Public review)]

Summary:

Asabuki and Clopath study stochastic sequence learning in recurrent networks of Poisson spiking neurons that obey Dale's law. Inspired by previous modeling studies, they introduce two distinct learning rules, to adapt excitatory-to-excitatory and inhibitory-to-excitatory synaptic connections. Through a series of computer experiments, the authors demonstrate that their networks can learn to generate stochastic sequential patterns, where states correspond to non-overlapping sets of neurons (cell assemblies) and the state-transition conditional probabilities are first-order Markov, i.e., the transition to a given next state only depends on the current state. Finally, the authors use their model to reproduce certain experimental songbird data involving highly-predictable and highly-uncertain transitions between song syllables. While the findings are only moderately surprising, this is a well-written and welcome detailed study that may be of interest to experts of plasticity and learning in recurrent neural networks that respect Dale's law.

Strengths:

This is an easy-to-follow, well-written paper, whose results are likely easy to reproduce. The experiments are clear and well-explained. In particular, the study of the interplay between excitation and inhibition (and their different plasticity rules) is a highlight of the study. The study of songbird experimental data is another good feature of this paper; finches are classical model animals for understanding sequence learning in the brain. I also liked the study of rapid task-switching, it's a good-to-know type of result that is not very common in sequence learning papers.

Weaknesses:

One weakness I see in this paper is the derivation of the learning rules, which is semi-heuristic. The paper studies Poisson spiking neurons, for which learning rules can be derived from a statistical objective, typically maximum likelihood, as previously done in the cited literature. The authors provide a brief section connecting the learning rules to gradient descent on objective functions, but the link is only heuristic or at least not entirely presented. The reason is that the neural network state is not fully determined by (or "clamped to") the target during learning (for instance, inhibitory neurons do not even have a target assigned). So, the (total) gradient should take into account the recurrent contributions from other neurons, and equation 13 does not appear to be complete/correct to me. Moreover, the target firing rate is a mixture of external currents with currents arising from other neurons in the recurrent network. The authors ideally should start from an actual distribution matching objective (e.g., KL divergence, and not such a squared error), so that their main claims immediately follow from the mathematical derivations. Along the same line, it would be excellent to get some additional insights on the interaction of the two distinct plasticity rules, one of the highlights of the study. This could be naturally achieved by relating their distinct rules to a common principled objective.

The other major weakness (albeit one that is clearly discussed by the authors) is that the study assumes that every excitatory neuron is directly given its target state when learning. In machine learning language, there are no 'hidden' excitatory neurons. While this assumption greatly simplifies the derivation of efficient and biologically-plausible learning rules that can be mapped to synaptic plasticity, it also limits considerably the distributions that can be learned by the network, more precisely to those that satisfy the Markov property.

---

## [Author Response]

The following is the authors’ response to the original reviews.

**Public Reviews:**

**Reviewer #1 (Public Review):**
In the presented manuscript, the authors investigate how neural networks can learn to replay presented sequences of activity. Their focus lies on the stochastic replay according to learned transition probabilities. They show that based on error-based excitatory and balance-based inhibitory plasticity networks can selforganize towards this goal. Finally, they demonstrate that these learning rules can recover experimental observations from song-bird song learning experiments.Overall, the study appears well-executed and coherent, and the presentation is very clear and helpful. However, it remains somewhat vague regarding the novelty. The authors could elaborate on the experimental and theoretical impact of the study, and also discuss how their results relate to those of Kappel et al, and others (e.g., Kappel et al (doi.org/10.1371/journal.pcbi.1003511)).

We agree with the reviewer that our previous manuscript lacked comparison with previously published similar works. While Kappel et al. demonstrated that STDP in winner-take-all circuits can approximate online learning of hidden Markov models (HMMs), a key distinction from our model is that their neural representations acquire deterministic sequential activations, rather than exhibiting stochastic transitions governing Markovian dynamics. Specifically, in their model, the neural representation of state B would be different in the sequences ABC and CBA, resulting in distinct deterministic representations like ABC and C'B'A', where ‘A’ and ‘A'’ are represented by different neural states (e.g., activations of different cell assemblies). In contrast, our network learns to generate stochastically transitioning cell assemblies which replay Markovian trajectories of spontaneous activity obeying the learned transition probabilities between neural representations of states. For example, starting from reactivation from assembly ‘A’, there may be an 80% probability to transition to assembly ‘B’ and 20% to ‘C’. Although Kappel et al.'s model successfully solves HMMs, their neural representations do not themselves stochastically transition between states according to the learned model. Similar to the Kappel et al.'s model, while the models proposed in Barber (2002) and Barber and Agakov (2002) learn the Markovian statistics, these models learned a static spatiotemporal input patterns only and how assemblies of neurons show stochastic transition in spontaneous activity has been still unclear. In contrast with these models, our model captures the probabilistic neural state trajectories, allowing spontaneous replay of experienced sequences with stochastic dynamics matching the learned environmental statistics.

We have included new sentences for explain these in ll. 509-533 in the revised manuscript.

Overall, the work could benefit if there was either (A) a formal analysis or derivation of the plasticity rules involved and a formal justification of the usefulness of the resulting (learned) neural dynamics;

We have included a derivation of our plasticity rules in ll. 630-670 in the revised manuscript. Consistent with our claim that excitatory plasticity updates the excitatory synapse to predict output firing rates, we have shown that the corresponding cost function measures the discrepancy between the recurrent prediction and the output firing rate. Similarly, for inhibitory plasticity, we defined the cost function that evaluates the difference between the excitatory and inhibitory potential within each neuron. We showed that the resulting inhibitory plasticity rule updates the inhibitory synapses to maintain the excitation-inhibition balance.

and/or (B) a clear connection of the employed plasticity rules to biological plasticity and clear testable experimental predictions. Thus, overall, this is a good work with some room for improvement.

Our proposed plasticity mechanism could be implemented through somatodendritic interactions. Analogous to previous computational works (Urbanczik and Senn, 2014; Asabuki and Fukai, 2020; Asabuki et al., 2022), our model suggests that somatic responses may encode the stimulus-evoked neural activity states, while dendrites encode predictions based on recurrent dynamics that aim to minimize the discrepancy between somatic and dendritic activity. To directly test this hypothesis, future experimental studies could simultaneously record from both somatic and dendritic compartments to investigate how they encode evoked responses and predictive signals during learning (Francioni et al., 2025).

We have included new sentences for explain these in ll. 476-484 in the revised manuscript.

**Reviewer #2 (Public Review):**
Summary:This work proposes a synaptic plasticity rule that explains the generation of learned stochastic dynamics during spontaneous activity. The proposed plasticity rule assumes that excitatory synapses seek to minimize the difference between the internal predicted activity and stimulus-evoked activity, and inhibitory synapses try to maintain the E-I balance by matching the excitatory activity. By implementing this plasticity rule in a spiking recurrent neural network, the authors show that the state-transition statistics of spontaneous excitatory activity agree with that of the learned stimulus patterns, which are reflected in the learned excitatory synaptic weights. The authors further demonstrate that inhibitory connections contribute to well-defined state transitions matching the transition patterns evoked by the stimulus. Finally, they show that this mechanism can be expanded to more complex state-transition structures including songbird neural data.Strengths:This study makes an important contribution to computational neuroscience, by proposing a possible synaptic plasticity mechanism underlying spontaneous generations of learned stochastic state-switching dynamics that are experimentally observed in the visual cortex and hippocampus. This work is also very clearly presented and well-written, and the authors conducted comprehensive simulations testing multiple hypotheses. Overall, I believe this is a well-conducted study providing interesting and novel aspects of the capacity of recurrent spiking neural networks with local synaptic plasticity.Weaknesses:This study is very well-thought-out and theoretically valuable to the neuroscience community, and I think the main weaknesses are in regard to how much biological realism is taken into account. For example, the proposed model assumes that only synapses targeting excitatory neurons are plastic, and uses an equal number of excitatory and inhibitory neurons.

We agree with the reviewer. The network shown in the previous manuscript consists of an equal number of excitatory and inhibitory neurons, which seems to lack biological plausibility. Therefore, we first tested whether a biologically plausible scenario would affect learning performance by setting the ratio of excitatory to inhibitory neurons to 80% and 20% (Supplementary Figure 7a; left). Even in such a scenario, the network still showed structured spontaneous activity (Supplementary Figure 7a; center), with transition statistics of replayed events matching the true transition probabilities (Supplementary Figure 7a; right). We then asked whether the model with our plasticity rule applied to all synapses would reproduce the corresponding stochastic transitions. We found that the network can learn transition statistics but only under certain conditions. The network showed only weak replay and failed to reproduce the appropriate transition (Supplementary Fig. 7b) if the inhibitory neurons were no longer driven by the synaptic currents reflecting the stimulus, due to a tight balance of excitatory and inhibitory currents on the inhibitory neurons. We then tested whether the network with all synapses plastic can learn transition statistics if the external inputs project to the inhibitory neurons as well. We found that, when each stimulus pattern activates a non-overlapping subset of neurons, the network does not exhibit the correct stochastic transition of assembly reactivation (Supplementary Fig. 7c). Interestingly, when each neuron's activity is triggered by multiple stimuli and has mixed selectivity, the reactivation reproduced the appropriate stochastic transitions (Supplementary Fig. 7d).

We have included these new results as new Supplementary Figure 7 and they are explained in ll.215-230 in the revised manuscript.

The model also assumes Markovian state dynamics while biological systems can depend more on history. This limitation, however, is acknowledged in the Discussion.

We have included the following sentence to provide a possible solution to this limitation: “Therefore, to learn higher-order stochastic transitions, recurrent neural networks like ours may need to integrate higher-order inputs with longer time scales.” in ll.557-559 in the revised manuscript.

Finally, to simulate spontaneous activity, the authors use a constant input of 0.3 throughout the study. Different amplitudes of constant input may correspond to different internal states, so it will be more convincing if the authors test the model with varying amplitudes of constant inputs.

We thank the reviewer for pointing this out. In the revised manuscript, we have tested constant input with three different strengths. If the strength is moderate, the network showed accurate encoding of transition statistics in the spontaneous activity as we have seen in Fig.2. We have additionally shown that the weaker background input causes spontaneous activity with lower replay rate, which in turn leads to high variance of encoded transition, while stronger inputs make assembly replay transitions more uniform. We have included these new results as new Supplementary Figure 6 and they are explained in ll.211214 in the revised manuscript.

**Reviewer #3 (Public Review):**
Summary:Asabuki and Clopath study stochastic sequence learning in recurrent networks of Poisson spiking neurons that obey Dale's law. Inspired by previous modeling studies, they introduce two distinct learning rules, to adapt excitatory-to-excitatory and inhibitory-to-excitatory synaptic connections. Through a series of computer experiments, the authors demonstrate that their networks can learn to generate stochastic sequential patterns, where states correspond to non-overlapping sets of neurons (cell assemblies) and the state-transition conditional probabilities are first-order Markov, i.e., the transition to a given next state only depends on the current state. Finally, the authors use their model to reproduce certain experimental songbird data involving highly-predictable and highly-uncertain transitions between song syllables.Strengths:This is an easy-to-follow, well-written paper, whose results are likely easy to reproduce. The experiments are clear and well-explained. The study of songbird experimental data is a good feature of this paper; finches are classical model animals for understanding sequence learning in the brain. I also liked the study of rapid task-switching, it's a good-to-know type of result that is not very common in sequence learning papers.Weaknesses:While the general subject of this paper is very interesting, I missed a clear main result. The paper focuses on a simple family of sequence learning problems that are well-understood, namely first-order Markov sequences and fully visible (nohidden-neuron) networks, studied extensively in prior work, including with spiking neurons. Thus, because the main results can be roughly summarized as examples of success, it is not entirely clear what the main point of the authors is.

We apologize the reviewer that our main claim was not clear. While various computational studies have suggested possible plasticity mechanisms for embedding evoked activity patterns or their probability structures into spontaneous activity (Litwin-Kumar et al., Nat. Commun. 2014, Asabuki and Fukai., Biorxiv 2023), how transition statistics of the environment are learned in spontaneous activity is still elusive and poorly understood. Furthermore, while several network models have been proposed to learn Markovian dynamics via synaptic plasticity (Brea, et al. (2013); Pfister et al. (2004); Kappel et al. (2014)), they have been limited in a sense that the learned network does not show stochastic transition in a neural state space. For instance, while Kappel et al. demonstrated that STDP in winner-take-all circuits can approximate online learning of hidden Markov models (HMMs), a key distinction from our model is that their neural representations acquire deterministic sequential activations, rather than exhibiting stochastic transitions governing Markovian dynamics. Specifically, in their model, the neural representation of state B would be different in the sequences ABC and CBA, resulting in distinct deterministic representations like ABC and C'B'A', where ‘A’ and ‘A'’ are represented by different neural states (e.g., activations of different cell assemblies). In contrast, our network learns to generate stochastically transitioning cell assemblies that replay Markovian trajectories of spontaneous activity obeying the learned transition probabilities between neural representations of states. For example, starting from reactivation from assembly ‘A’, there may be an 80% probability to transition to assembly ‘B’ and 20% to ‘C’. Although Kappel et al.'s model successfully solves HMMs, their neural representations do not themselves stochastically transition between states according to the learned model. Similar to the Kappel et al.'s model, while the models proposed in Barber (2002) and Barber and Agakov (2002) learn the Markovian statistics, these models learned a static spatiotemporal input patterns only and how assemblies of neurons show stochastic transition in spontaneous activity has been still unclear. In contrast with these models, our model captures the probabilistic neural state trajectories, allowing spontaneous replay of experienced sequences with stochastic dynamics matching the learned environmental statistics.

We have explained this point in ll.509-533 in the revised manuscript.

Going into more detail, the first major weakness I see in this paper is the heuristic choice of learning rules. The paper studies Poisson spiking neurons (I return to this point below), for which learning rules can be derived from a statistical objective, typically maximum likelihood. For fully-visible networks, these rules take a simple form, similar in many ways to the E-to-E rule introduced by the authors. This more principled route provides quite a lot of additional understanding on what is to be expected from the learning process.

We thank the reviewer for pointing this out. To better demonstrate the function of our plasticity rules, we have included the derivation of the rules of synaptic plasticity in ll. 630-670 in the revised manuscript. Consistent with our claim that excitatory plasticity updates the excitatory synapse to predict output firing rates, we have shown that the corresponding cost function measures the discrepancy between the recurrent prediction and the output firing rate. Similarly, for inhibitory plasticity, we defined the cost function that evaluates the difference between the excitatory and inhibitory potential within each neuron. We showed that the resulting inhibitory plasticity rule updates the inhibitory synapses to maintain the excitation-inhibition balance.

For instance, should maximum likelihood learning succeed, it is not surprising that the statistics of the training sequence distribution are reproduced. Moreover, given that the networks are fully visible, I think that the maximum likelihood objective is a convex function of the weights, which then gives hope that the learning rule does succeed. And so on. This sort of learning rule has been studied in a series of papers by David Barber and colleagues [refs. 1, 2 below], who applied them to essentially the same problem of reproducing sequence statistics in recurrent fully-visible nets. It seems to me that one key difference is that the authors consider separate E and I populations, and find the need to introduce a balancing I-to-E learning rule.

The reviewer’s understanding that inhibitory plasticity to maintain EI balance is one of a critical difference from previous works is correct. However, we believe that the most striking point of our study is that we have shown numerically that predictive plasticity rules enable recurrent networks to learn and replay the assembly activations whose transition statistics match those of the evoked activity. Please see our reply above.

Because the rules here are heuristic, a number of questions come to mind. Why these rules and not others - especially, as the authors do not discuss in detail how they could be implemented through biophysical mechanisms? When does learning succeed or fail? What is the main point being conveyed, and what is the contribution on top of the work of e.g. Barber, Brea, et al. (2013), or Pfister et al. (2004)?

Our proposed plasticity mechanism could be implemented through somatodendritic interactions. Analogous to previous computational works (Senn, Asabuki), our model suggests that somatic responses may encode the stimulusevoked neural activity states, while dendrites encode predictions based on recurrent dynamics that aim to minimize the discrepancy between somatic and dendritic activity. To directly test this hypothesis, future experimental studies could simultaneously record from both somatic and dendritic compartments to investigate how they encode evoked responses and predictive signals during learning.

To address the point of the reviewer, we conducted addionnal simulations to test where the model fails. We found that the model with our plasticity rule applied to all synapses only showed faint replays and failed to replay the appropriate transition (Supplementary Fig. 7b). This result is reasonable because the inhibitory neurons were no longer driven by the synaptic currents reflecting the stimulus, due to a tight balance of excitatory and inhibitory currents on the inhibitory neurons. Our model predicts that mixed selectivity in the inhibitory population is crucial to learn an appropriate transition statistics (Supplementary Fig. 7d). Future work should clarify the role of synaptic plasticity on inhibitory neurons, especially plasticity at I to I synapses. We have explained this result as new supplementary Figure7 in the revised manuscript.

The use of a Poisson spiking neuron model is the second major weakness of the study. A chief challenge in much of the cited work is to generate stochastic transitions from recurrent networks of deterministic neurons. The task the authors set out to do is much easier with stochastic neurons; it is reasonable that the network succeeds in reproducing Markovian sequences, given an appropriate learning rule. I believe that the main point comes from mapping abstract Markov states to assemblies of neurons. If I am right, I missed more analyses on this point, for instance on the impact that varying cell assembly size would have on the findings reported by the authors.

The reviewer’s understanding is correct. Our main point comes from mapping Markov statistics to replays of cell assemblies. In the revised manuscript, we performed additional simulations to ask whether varying the size of the cell assemblies would affect learning. We ran simulations with two different configurations in the task shown in Figure 2. The first configuration used three assemblies with a size ratio of 1:1.5:2. After training, these assemblies exhibited transition statistics that closely matched those of the evoked activity (Supplementary Fig.4a,b). In contrast, the second configuration, which used a size ratio of 1:2:3, showed worse performance compared to the 1:1.5:2 case (Supplementary Fig.4c,d). These results suggest that the model can learn appropriate transition statistics as long as the size ratio of the assemblies is not drastically varied.

Finally, it was not entirely clear to me what the main fundamental point in the HVC data section was. Can the findings be roughly explained as follows: if we map syllables to cell assemblies, for high-uncertainty syllable-to-syllable transitions, it becomes harder to predict future neural activity? In other words, is the main point that the HVC encodes syllables by cell assemblies?

The reviewer's understanding is correct. We wanted to show that if the HVC learns transition statistics as a replay of cell assemblies, a high-uncertainty syllable-to-syllable transition would make predicting future reactivations more difficult, since trial-averaged activities (i.e., poststimulus activities; PSAs) marginalized all possible transitions in the transition diagram.

(1) Learning in Spiking Neural Assemblies, David Barber, 2002. URL: https://proceedings.neurips.cc/paper/2002/file/619205da514e83f869515c782a328d3c-Paper.pdf

(2) Correlated sequence learning in a network of spiking neurons usingmaximum likelihood, David Barber, Felix Agakov, 2002. URL: http://web4.cs.ucl.ac.uk/staff/D.Barber/publications/barber-agakovTR0149.pdf

**Recommendations for the authors:**

**Reviewer #1 (Recommendations For The Authors):**
In more detail:A) Theoretical analysisThe plasticity rules in the study are introduced with a vague reference to previous theoretical studies of others. Doing this, one does not provide any formal insight as to why these plasticity rules should enable one to learn to solve the intended task, and whether they are optimal in some respect. This becomes noticeable, especially in the discussion of the importance of inhibitory balance, which does not go into any detail, but rather only states that its required, both in the results and discussion sections. Another unclarity appears when error-based learning is discussed and compared to Hebbian plasticity, which, as you state, "alone is insufficient to learn transition probabilities". It is not evident how this claim is warranted, nor why error-based plasticity in comparison should be able to perform this (other than referring to the simulation results). Please either clarify formally (or at least intuitively) how plasticity rules result in the mentioned behavior, or alternatively acknowledge explicitly the (current) lack of intuition.The lack of formal discussion is a relevant shortcoming compared to previous research that showed very similar results with formally more rigorous and principled approaches. In particular, Kappel et al derived explicitly how neural networks can learn to sample from HMMs using STDP and winner-take-all dynamics. Even though this study has limitations, the relation with respect to that work should be made very clear; potentially the claims of novelty of some results (sampling) should be adjusted accordingly. See also Yanping Huang, Rajesh PN Rao (NIPS 2014), and possibly other publications. While it might be difficult to formally justify the learning rules post-hoc, it would be very helpful to the field if you very clearly related your work to that of others, where learning rules have been formally justified, and elaborate on the intuition of how the employed rules operate and interact (especially for inhibition).Lastly, while the importance of sampling learned transition probabilities is discussed, the discussion again remains on a vague level, characterized by the lack of references in the relevant paragraphs. Ideally, there should be a proof of concept or a formal understanding of how the learned behaviour enables to solve a problem that is not solved by deterministic networks. Please incorporate also the relation to the literature on neural sampling/planning/RL etc. and substantiate the claims with citations.

We have included sentences in ll. 691-696 in the revised manuscript to explain that for Poisson spiking neurons, the derived learning rule is equivalent to the one that minimizes the Kullback-Leibler divergence between the distributions of output firing and the dendritic prediction, in our case, the recurrent prediction (Asabuki and Fukai; 2020). Thus, the rule suggests that the recurrent prediction learns the statistical model of the evoked activity, which in turn allows the network to reproduce the learned transition statistics.

We have also added a paragraph to discuss the differences between previously published similar models (e.g., Kappel et al.). Please see our response above.

B) Connection to biologyThe plasticity rules in the study are introduced with a vague reference to previous theoretical studies of others. Please discuss in more detail if these rules (especially the error-based learning rule) could be implemented biologically and how this could be achieved. Are there connections to biologically observed plasticity? E.g. for error-based plasticity has been discussed in the original publication by Urbanzcik and Senn, or more recently by Mikulasch et al (TINS 2023). The biological plausibility of inhibitory balance has been discussed many times before, e.g. by Vogels and others, and a citation would acknowledge that earlier work. This also leaves the question of how neurons in the songbird experiment could adapt and if the model does capture this well (i.e., do they exhibit E-I balance? etc), which might be discussed as well.Last, please provide some testable experimental predictions. By proposing an interesting experimental prediction, the model could become considerably more relevant to experimentalists. Also, are there potentially alternative models of stochastic sequence learning (e.g., Kappel et al)? How could they be distinguished? (especially, again, why not Hebbian/STDP learning?)

We have cited the Vogels paper to acknowledge the earlier work. We have also included additional paragraphs to discuss a possible biologically plausible implementation of our model and how our model differs from similar models proposed previously (e.g., Kappel et al.). Please see our response above.

Other commentsAs mentioned, a derivation of recurrent plasticity rules is missing, and parameters are chosen ad-hoc. This leaves the question of how much the results rely on the specific choice of parameters, and how robust they are to perturbations. As a robustness check, please clarify how the duration of the Markov states influences performance. It can be expected that this interacts with the timescale of recurrent connections, so having longer or shorter Markov states, as it would be in reality, should make a difference in learning that should be tested and discussed.

We thank the reviewer for pointing this out. To address this point, we performed new simulations and asked to what extent the duration of Markov states affect performance. Interestingly, even when the network was trained with input states of half the duration, the distributions of the durations of assembly reactivations remain almost identical to those in the original case (Supplementary Figure 3a). Furthermore, the transition probabilities in the replay were still consistent with the true transition probabilities (Supplementary Figure 3b). We have also included the derivation of our plasticity rule in ll. 630-670 in the revised manuscript.

Similarly, inhibitory plasticity operates with the same plasticity timescale parameter as excitatory plasticity, but, as the authors discuss, lags behind excitatory plasticity in simulation as in experiment. Is this required or was the parameter chosen such that this behaviour emerges? Please clarify this in the methods section; moreover, it would be good to test if the same results appear with fast inhibitory plasticity.

We have performed a new simulation and showed that even when the learning rate of inhibitory plasticity was larger than that of excitatory plasticity, inhibitory plasticity still occurred on a slower timescale than excitatory plasticity. We have included this result in a new Supplementary Figure 2 in the revised manuscript.

What is the justification (biologically and theoretically) for the memory trace h and its impact on neural spiking? Is it required for the results or can it be left away? Since this seems to be an important and unconventional component of the model, please discuss it in more detail.

In the model, it is assumed that each stimulus presentation drives a specific subset of network neurons with a fixed input strength, which avoids convergence to trivial solutions. Nevertheless, we choose to add this dynamic sigmoid function to facilitate stable replay by regulating neuron activity to prevent saturation. We have explained this point in ll.605-611 in the revised manuscript.

**Reviewer #2 (Recommendations For The Authors):**
I noticed a couple of minor typos:Page 3 "underly"->"underlie"Page 7 "assemblies decreased settled"->"assemblies decreased and settled"

We have modified the text. We thank the reviewer for their careful review.

I think Figure 1C is rather confusing and not intuitive.

We apologize that the Figure 1C was confusing. In the revised figure, we have emphasized the flow of excitatory and inhibitory error for updating synapses.

**Reviewer #3 (Recommendations For The Authors):**
One possible path to improve the paper would be to establish a relationship between the proposed learning rules and e.g. the ones derived by Barber.When reading the paper, I was left with a number of more detailed questions I omitted from the public review:(1) The authors introduce a dynamic sigmoidal function for excitatory neurons, Eq. 3. This point requires more discussion and analysis. How does this impact the results?

In the model, it is assumed that each stimulus presentation drives a specific subset of network neurons with a fixed input strength, which avoids convergence to trivial solutions. Nevertheless, we choose to add this dynamic sigmoid function to facilitate stable replay by regulating neuron activity to prevent saturation. We have explained this point in ll.605-611 in the revised manuscript.

(2) For Poisson spiking neurons, it would be great to understand what cell assemblies bring (apart from biological realism, i.e., reproducing data where assemblies can be found), compared to self-connected single neurons. For example, how do the results shown in Figure 2 depend on assembly size?

We have changed the cell assembly size ratio and how it affects learning performance in a new Supplementary Figure 4. Please see our reply above.

(3) The authors focus on modeling spontaneous transitions, corresponding to a highly stochastic generative model (with most transition probabilities far from 1). A complementary question is that of learning to produce a set of stereotypical sequences, with probabilities close to 1. I wondered whether the learning rules and architecture of the model (in particular under the I-to-E rule) would also work in such a scenario.

We thank the reviewer for pointing this out. In fact, we had the same question, so we considered a situation in which the setting in Figure 2 includes both cases where the transition matrix is very stochastic (prob=0.5) and near deterministic (prob=0.9).

(4) An analysis of what controls the time so that the network stays in a certain state would be welcome.

We trained the network model in two cases, one with a fast speed of plasticity and one with a slow speed of plasticity. As a result, we found that the duration of assembly becomes longer in the slow learning case than in the fast case. We have included these results as Supplementary Figure 5 in the revised manuscript.

Regarding the presentation, given that this is a computational modeling paper, I wonder whether *all* the formulas belong in the Methods section. I found myself skipping back and forth to understand what the main text meant, mainly because I missed a few key equations. I understand that this is a style issue that is very much community-dependent, but I think readability would improve drastically if the main model and learning rule equations could be introduced in the main text, as they start being discussed.

We thank the reviewer for the suggestion. To cater to a wider audience, we try to explain the principle of the paper without using mathematical formulas as much as possible in the main text.